# Enhanced Red Emission from Amorphous Silicon Carbide Films via Nitrogen Doping

**DOI:** 10.3390/mi13122043

**Published:** 2022-11-22

**Authors:** Guangxu Chen, Sibin Chen, Zewen Lin, Rui Huang, Yanqing Guo

**Affiliations:** School of Materials Science and Engineering, Hanshan Normal University, Chaozhou 521041, China

**Keywords:** photoluminescence, optical properties, nitrogen, SiCx

## Abstract

The enhanced red photoluminescence (PL) from Si-rich amorphous silicon carbide (a-SiC_x_) films was analyzed in this study using nitrogen doping. The increase in nitrogen doping concentration in films results in the significant enhancement of PL intensity by more than three times. The structure and bonding configuration of films were investigated using Raman and Fourier transform infrared absorption spectroscopies, respectively. The PL and analysis results of bonding configurations of films suggested that the enhancement of red PL is mainly caused by the reduction in nonradiative recombination centers as a result of the weak Si–Si bonds substituted by Si–N bonds.

## 1. Introduction

Efficient Si-based light sources are indispensable components for realizing Si-based monolithic optoelectronic integrated circuits. Silicon-based materials, such as silicon oxide (SiO_x_), silicon nitride (SiN_x_), silicon carbine (SiC_x_), and silicon oxycarbide (SiC_x_O_y_), have been extensively investigated over the past decade to obtain efficient Si-based light sources [1,2,3,4,5,6,7,8,9,10]. However, light emission efficiency is still very low and commercial applications are difficult to meet. The low emission efficiency is mainly limited by the strongly nonradiative recombination originating from defects and low carrier injection efficiency [11]. Compared with the wide bandgap of SiO_2_ and SiN_x_, SiC_x_ features a smaller bandgap, which is more conducive to carrier injection and achieving electroluminescence at lower driving voltages in SiC_x_-based light-emitting devices [5,12]. Although SiC_x_ possesses these advantages, research of its luminescence is still progressing slowly due to the strong nonradiative recombination derived from band tails and defect states in SiC_x_ induced by its structural disorder. In order to reduce nonradiative recombination, hydrogen treatment is often used to passivate defective states and improve structural order by etching weak Si–Si bonds [13]. However, the hydrogen used for passivation will fall off at high temperatures and lead to the recovery of defective states. Therefore, the regulation of defective states has become a key process for obtaining efficient SiC-based devices with suppressed nonradiative recombination centers. In previous research, the passivation of nitrogen on the Si nanocrystals surface was found to effectively increase the probability of radiative recombination [14]. Li et al. reported surface nitrogen-capped Si NPs with PL efficiency up to 90% at wavelength of 560 nm. However, up to now, there is no report on the effect of nitrogen passivation on the luminescent properties of SiC_x_ films [14].

In this study, the red luminescent Si-rich a-SiC_x_ films doped with N were fabricated using very high-frequency plasma-enhanced chemical vapor deposition (VHF-PECVD). The increase in nitrogen doping concentration in films significantly improves red light emission by more than three times. The enhanced red PL is discussed, which is mainly caused by the reduction in nonradiative recombination centers as a result of the weak Si–Si bonds substituted by Si–N bonds.

## 2. Experimental Details

Amorphous Si-rich a-SiC_x_ films doped with N were prepared via very high frequency plasma-enhanced chemical vapor deposition using SiH_4_, CH_4_, and NH_3_ as reaction gas sources. These fabrications were carried out at a temperature of 250 °C. The RF power and the deposition pressure for the growth were kept at 30 W and 20 Pa, respectively. Flow rates of SiH_4_ and CH_4_ were maintained at 2.5 and 5 sccm (standard cubic centimeter per minute), respectively, while those of NH_3_ varied from 0 sccm to 2 sccm. The PL spectra of the films were recorded by an Edinburgh FLS1000 fluorescence spectrometer equipped with a 450 W steady Xe lamp. The PL decay curves were measured by an Edinburgh FLS1000 spectrometer using a 372 nm picosecond laser (pulse width 44 ps, repetition rate = 20 MHz). The absorption spectra of the films were obtained with a Shimadzu UV-3600 spectrophotometer (Shimadzu UV-3600, Shimadzu Corporation, Kyoto, Japan). Microstructures of the films were evaluated using a Horiba LabRAM HR Evolution Raman spectrometer. Bonding structures were recorded via Fourier transform infrared absorption (FTIR) spectrometry (Shimadzu IR Pretige-21).

## 3. Results and Discussion

Figure 1 shows the PL spectra of thin films prepared at different NH_3_ flow rates under an excitation wavelength of 325 nm. The prepared film without NH_3_ only shows weak red light emission that peaks at ~780 nm. The emission peak position of the film changes minimally but the red light emission gradually intensifies with the addition of NH_3_. The red light emission of the film is nearly three times stronger than that of the film prepared without NH_3_ when the NH_3_ flow rate increases to 2 sccm. This phenomenon is consistent with improvement of the photoluminescence properties in a-SiN_x_ films by the introduction of hydrogen [15]. Figure 2a,b reveal that the change in emission peak position and the full width at half maximum with the variation of the excitation wavelength is insignificant. This finding clearly features defect luminescence characteristics similar to those observed in defect-related luminescent Si-based materials, such as SiC_x_ and SiC_x_O_y_ [6,8].

The microstructure of films was characterized using Raman scattering spectra to clarify the luminescence enhancement (Figure 3a). The Raman spectra show the typical features of a-Si vibration modes. The Raman peaks at around 150 cm^−1^ and 480 cm^−1^ are attributed to transverse acoustic (TA) and transverse optical (TO) phonon frequencies, respectively [16,17]. This finding indicates that amorphous silicon clusters exist but Si and SiC nanocrystals are absent in the films. Moreover, with an increased NH_3_ flow rate, one can see that the intensity ratio of the TA mode (150 cm^−1^) to the TO mode (480 cm^−1^) tends to decrease. These observations suggest a reduced short-range and medium-range disorder of the Si–Si_4_ network [17]. This phenomenon is closely related to the increase in N content in the film. The weak Si–Si bond in the film will be gradually etched and replaced by the Si–N bond with the addition of N given that the Si–N bond energy (355 kJ/mol) is greater than the Si–Si bond energy (222 kJ/mol). It seems that the reduction in weak Si–Si bonds is responsible for the significant enhancement in red PL [18]. The surface morphology of the films was further revealed by atomic force microscopy (AFM) as shown in Figure 3b,c. The average RMS values for the films are around 20 nm. No obvious change can be observed in the surface morphology between the films prepared by different NH_3_ flow rates. This ruled out the possibility that the enhanced PL was caused by the increasing light extraction from the films.

Films were analyzed with an FTIR spectrometer to further explore the bonding configuration of films prepared with different NH_3_ flow rates. The results are shown in Figure 4. The FTIR absorption spectrum for the film prepared without NH_3_ mainly displays the following vibrational bands [6,16]: the absorption band at 640 cm^−1^ corresponds to the rocking vibration mode of SiH_n_, the 780 cm^−1^ band is related to the stretching vibration mode of Si–C, the 1250 cm^−1^ band corresponds to the stretching vibration mode of Si–C, and the 2100 cm^−1^ band is attributed to the stretching vibration mode of H–Si–Si_3_. The FTIR absorption spectrum clearly showed the feature of Si-rich SiC_x_. One can see that the rocking vibration mode of SiH_n_ at 640 cm^−1^ and the stretching vibration mode of H–Si–Si_3_ at 2100 cm^−1^ gradually weaken with the increase in NH_3_ flow rates. In contrast, the stretching vibration mode of Si–N appears and becomes intense with the addition of NH_3_. With the NH_3_ flow rate increasing from 0.5 to 2 sccm, the density of Si–N bond is estimated to be increased from 0.2 × 10^22^ cm^−2^ to 0.7 × 10^22^ according to the following equation [19]:(1)$$N=A\int \frac{\alpha \left(\omega \right)}{\omega }d\omega \;$$
where *α*(*ω*) is the absorption coefficient, *ω* is the wave number of the corresponding absorption band, and A is equal to 6.3 × 10^18^ cm^−2^, which is related to the absorption cross-section of the Si–N vibration mode. This finding strongly indicates that the weak Si–Si bond is gradually replaced by the Si–N bond with the continuous incorporation of N.

Figure 5 shows the PL decay traces of films prepared under different NH_3_ flow rates. The decay process in each case can be properly fitted with a biexponential function as follows:(2)$$I(t)=I_{0}{+A}_{1}\exp(\frac{-t}{\tau_{1}})+A_{2}\exp(\frac{-t}{\tau_{2}})$$
where *I*_0_ is the background level; *τ*_1_ and *τ*_2_ are the weight fraction and lifetime of each exponential decay component, respectively; and *A*_1_ and *A*_2_ are the corresponding amplitudes [20]. Thus, the average lifetime τ can be estimated as follows [20]:*τ* = (*A*_1_ × *τ*_1_^2^ + *A*_2_ × *τ*_2_^2^)/(*A*_1_ × *τ*_1_ + *A*_2_ × *τ*_2_)(3)

Figure 5 shows that all measured films feature a fast dynamic decay with lifetimes of nanoseconds. Moreover, the PL lifetime gradually increases from 3.2 ns to 3.8 ns with the increase in NH_3_ flow rates. The comparison of Figure 1 and Figure 5 shows that the evolution of the PL lifetime with NH_3_ flow rates is the same as that of PL intensity with NH_3_ flow rates. This finding strongly indicates that the improved red light emission is due to the reduction in nonradiative recombination centers in the film. As demonstrated by Raman spectra in Figure 3, the amorphous silicon component in the film gradually decreases with the increase in the NH_3_ flow rate. Meanwhile, the Si–N bond density increases with the increase in the NH_3_ flow rate (Figure 4). These results clearly demonstrate that some Si–Si bonds are replaced by Si–N bonds. The weak Si–Si bond in the film will likely be gradually etched and replaced by the Si–N bond with the addition of N, given that the Si–N bond energy (355 kJ/mol) is greater than the Si–Si bond energy (222 kJ/mol). The addition of nitrogen evidently reduces the nonradiative recombination centers in the film. Therefore, the increase in NH_3_ flow rate increases the PL lifetime and significantly enhances red light emissions.

## 4. Conclusions

Red luminescent Si-rich a-SiC_x_ films doped with N were fabricated using VHF-PECVD. The increase in nitrogen doping concentration in the films significantly improves red light emissions by more than three times. The PL results and analyses of bonding configurations of films demonstrated that the significant enhancement in red PL is caused by the effective reduction in nonradiative recombination centers from the reduction in weak Si–Si bonds substituted with Si–N bonds.

## Figures and Tables

**Figure 1 micromachines-13-02043-f001:**
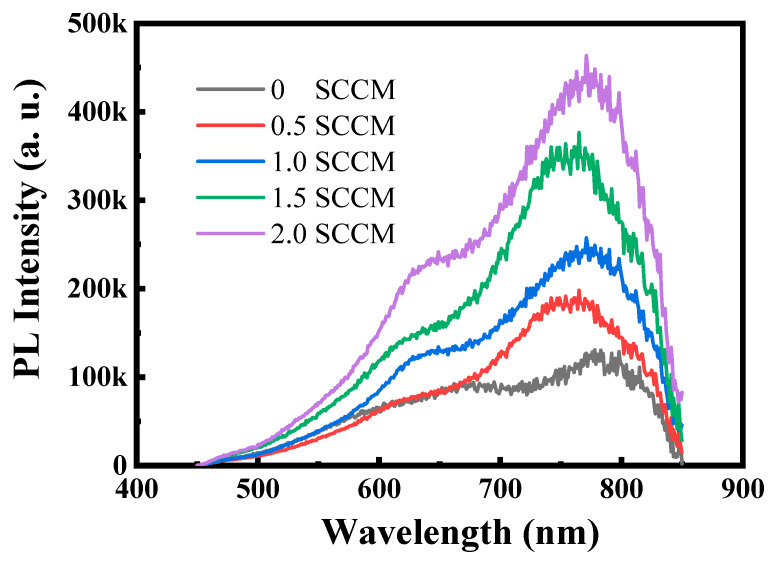
PL spectra of thin films prepared at different NH_3_ flow rates under an excitation wavelength of 325 nm.

**Figure 2 micromachines-13-02043-f002:**
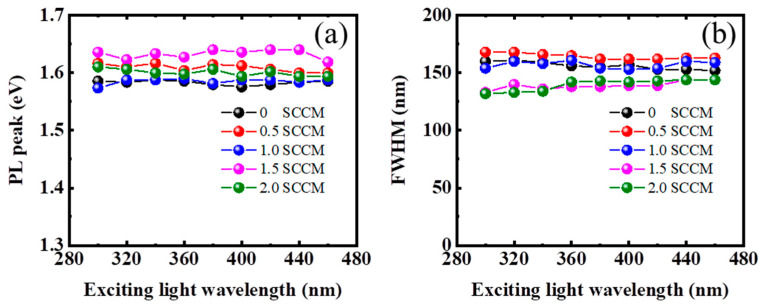
Emission peak position (**a**) and the full width at half maximum (**b**) with the variation of the excitation wavelength for the films prepared at different NH_3_ flow rates, respectively.

**Figure 3 micromachines-13-02043-f003:**
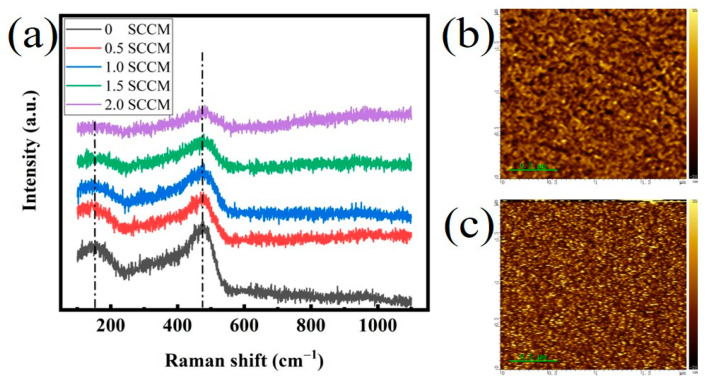
(**a**) Raman scattering spectra of the films prepared at different NH_3_ flow rates, (**b**,**c**) atomic force microscopic images of the films prepared by different NH_3_ flow rates of 1 sccm and 2 sccm, respectively.

**Figure 4 micromachines-13-02043-f004:**
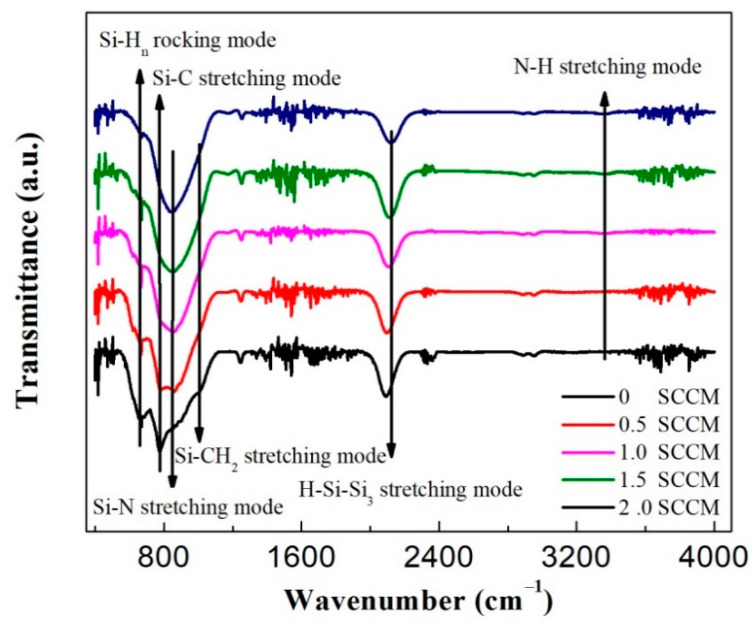
FTIR absorption spectra of the films prepared at different NH_3_ flow rates.

**Figure 5 micromachines-13-02043-f005:**
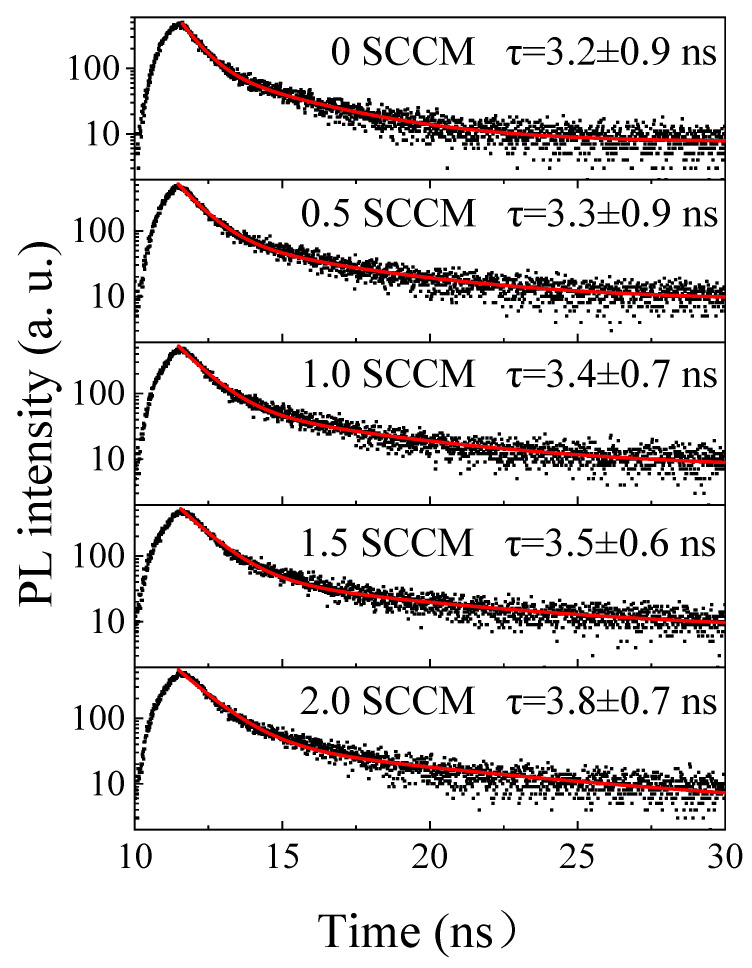
PL decay traces of the films prepared at different NH_3_ flow rates.

## Data Availability

Data underlying the results presented in this paper are not publicly available at this time but may be obtained from the authors upon reasonable request.

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
