# Peer review of "Enhanced Red Emission from Amorphous Silicon Carbide Films via Nitrogen Doping"

_micromachines, 2022, doi:10.3390/mi13122043_

Round 1

Reviewer 1 Report (New Reviewer)

This manuscript reports on the enhancement of the red emission through the nitrogen doping of SiC, which is a good idea.The manuscript is well written in details and the authors presented very well their findings. Thus, I recommand this work for publication.

Author Response

We sincerely appreciate the Editor’s comments and suggestions about our manuscript. According to your requirements, we have carefully corrected the technical issues in the revised manuscript. Following is our point-by-point reply.

Reviewer: 1

This manuscript reports on the enhancement of the red emission through the nitrogen doping of SiC, which is a good idea.The manuscript is well written in details and the authors presented very well their findings. Thus, I recommand this work for publication.

Response:

We sincerely thank the reviewer for his/her positive comments.

Reviewer 2 Report (New Reviewer)

Comments to the Authors:

This article has interesting results from the incorporation of Nitrogen in the a-Si C network. They found an enhanced red PL signal from Si-rich a-SiCx films through nitrogen doping. Here, the authors showed improved PL intensity as the nitrogen concentration in films increased.

Also, the bonding configuration of films was investigated using Raman and FTIR absorption.

However, there are some details to be elucidated to complete the article.

1. The experiment was done using NH3 flow from 0 to 2.0 sccm.

As the NH3 flow, and so the nitrogen content increases in the film, the resulting PL increases. Why the authors do not give a piece of information, what is the resulting PL with further NH3 flow increment?

Give a comment about this pending process.

2. The authors mentioned that the weak Si–Si bond in the film will be gradually etched and replaced by the Si–N bond being Si–N bond energy of 355 kJ/mol, and with a Si–Si bond energy of 222 kJ/mol.

But how the authors could trust that the enhancement in red PL is due to the reduction of “weak Si–Si bonds”. Please give a shred of evidence or proper references.

Also, it is suggested to enlarge Fig. 4, to let observe closely Si-N, or Si-related bonding tendencies.

Author Response

Response to the comments of Editor and Reviewer

We sincerely appreciate the Editor’s comments and suggestions about our manuscript. According to your requirements, we have carefully corrected the technical issues in the revised manuscript. Following is our point-by-point reply.

 Reviewer: 2

This article has interesting results from the incorporation of Nitrogen in the a-Si C network. They found an enhanced red PL signal from Si-rich a-SiCx films through nitrogen doping. Here, the authors showed improved PL intensity as the nitrogen concentration in films increased. Also, the bonding configuration of films was investigated using Raman and FTIR absorption.However, there are some details to be elucidated to complete the article.

  • (1)The experiment was done using NH3 flow from 0 to 2.0 sccm. As the NH3 flow, and so the nitrogen content increases in the film, the resulting PL increases. Why the authors do not give a piece of information, what is the resulting PL with further NH3 flow increment? Give a comment about this pending process.

Response:

In previous work, we have studied this work. With further increasing NH3 flow rate from 2.0 to 15 sccm, the structure of the films gradually changes from silicon carbide phase to silicon nitride phase. Although the PL intensity increases with further increasing NH3 flow rate, the PL band blueshifts from the near-infrared to the blue region, and the optical bandgap increased from 2.3 eV to 5.0 eV. This work has been published in Micromachines 202112(6), 637.

  • (2)The authors mentioned that the weak Si–Si bond in the film will be gradually etched and replaced by the Si–N bond being Si–N bond energy of 355 kJ/mol, and with a Si–Si bond energy of 222 kJ/mol. But how the authors could trust that the enhancement in red PL is due to the reduction of “weak Si–Si bonds”. Please give a shred of evidence or proper references. also, it is suggested to enlarge Fig. 4, to let observe closely Si-N, or Si-related bonding tendencies.

Response:

According to the Reviewer’s suggestion, we have added the corresponding reference and enlarge Fig. 4 in the revised manuscript. Please see Page 9, Reference 18 and Page 5, Figure 4.

[1] Rui, Y.; Chen, D.; Xu, J.; et al., Hydrogen-induced recovery of photoluminescence from annealed a‐Si:H∕a‐SiO2 multilayers. J. Appl. Phys. 2005, 98, 033532-033532-6.

Reviewer 3 Report (New Reviewer)

The manuscript titled “Enhanced red emission from amorphous silicon carbide films via nitrogen doping” The author prepared Si-rich amorphous silicon carbide (a- SiCx) films. The PL and FT-IR and Raman analysis integrate that the enhancement of red PL is mainly due to the reduction of nonradiative recombination centers as a result of the weak Si–Si bonds replaced by Si–N bonds.

·            the abstract is well written.

·               Regarding the introduction, the authors should add the importance of nitrogen doping after discussing the hydrogen treatment and its drawbacks.

·               For results and discussion part: Data are clear and interesting finding; you should compare your results with other in literature survey.

Author Response

Response to the comments of Editor and Reviewer

We sincerely appreciate the Editor’s comments and suggestions about our manuscript. According to your requirements, we have carefully corrected the technical issues in the revised manuscript. Following is our point-by-point reply.

Reviewer: 3

The manuscript titled “Enhanced red emission from amorphous silicon carbide films via nitrogen doping” The author prepared Si-rich amorphous silicon carbide (a- SiCx) films. The PL and FT-IR and Raman analysis integrate that the enhancement of red PL is mainly due to the reduction of nonradiative recombination centers as a result of the weak Si–Si bonds replaced by Si–N bonds.  the abstract is well written.

1)   Regarding the introduction, the authors should add the importance of nitrogen doping after discussing the hydrogen treatment and its drawbacks.

Response:

 We have added the corresponding information in the revised manuscript. Please see Page 2, line 3-6.

2)   For results and discussion part: Data are clear and interesting finding; you should compare your results with other in literature survey.

 Response:

We sincerely thank the reviewer for his/her positive comments. We have added the corresponding information in the revised manuscript. Please see Page 3, line 7-12.

Reviewer 4 Report (New Reviewer)

Review

“Enhanced red emission from amorphous silicon carbide films 2

via nitrogen doping”

by

Guangxu Chen, Sibin Chen, Zewen Lin, Rui Huang and Yanqing Guo.

       The aim of this work is to increase the intensity of the red photoluminescence (PL) spectrum of amorphous silicon carbide films enriched with silicon. As a result of experimental study, the authors found that adding NH3 to the mixture of CH4 and SiH4 gases in the process of silicon carbide production the intensity of the red photoluminescence spectrum increases significantly. The PL attenuation time also increases.

In my opinion, the work is very interesting. It obtained new results, which are confirmed by reliable experimental data. 

There are several questions. I would like the authors to clarify them.

1. It is well known that the chemical reaction between ammonia and methane occurs in the presence of a platinum catalyst at high temperature, about 1300C. The temperature at which the authors work is much lower. However, they use a plasma source to produce silicon carbide. The question is as follows. Does the reaction between CH4 and NH3 not occur in the plasma at 250C?

2.    Does not the reaction between NH3 and SiH4 occur in the gas phase?

Overall, I believe that the article, after clarifying these points, can be accepted for publication in the journal Micromachines.

Author Response

Response to the comments of Editor and Reviewer

We sincerely appreciate the Editor’s comments and suggestions about our manuscript. According to your requirements, we have carefully corrected the technical issues in the revised manuscript. Following is our point-by-point reply.

Reviewer: 4

 The aim of this work is to increase the intensity of the red photoluminescence (PL) spectrum of amorphous silicon carbide films enriched with silicon. As a result of experimental study, the authors found that adding NH3 to the mixture of CH4 and SiH4 gases in the process of silicon carbide production the intensity of the red photoluminescence spectrum increases significantly. The PL attenuation time also increases.In my opinion, the work is very interesting. It obtained new results, which are confirmed by reliable experimental data. There are several questions. I would like the authors to clarify them. Overall, I believe that the article, after clarifying these points, can be accepted for publication in the journal Micromachines.

  • It is well known that the chemical reaction between ammonia and methane occurs in the presence of a platinum catalyst at high temperature, about 1300C. The temperature at which the authors work is much lower. However, they use a plasma source to produce silicon carbide. The question is as follows. Does the reaction between CH4 and NH3 not occur in the plasma at 250C?

Response:

CH4 can reacts with NH3 in the plasma at 250C [1-2]

[1] M. Othman, et al, Effects of hydrogen dilution on CNx film properties deposited using rf PECVD from a mixture of ethane, nitrogen and hydrogen. Materials Chemistry and Physics 2014, 144, 377-384.

[2] F. Černý, et al, Properties of CNx films prepared by PECVD. Diamond and Related Materials 1999, 8, 1730-1731.

2)  Does not the reaction between NH3 and SiH4 occur in the gas phase?

Response:

The reaction between NH3 and SiH4 mostly occurs on the substrates.

Round 2

Reviewer 2 Report (New Reviewer)

The article has been imporved.

Probably is ready to be edited

This manuscript is a resubmission of an earlier submission. The following is a list of the peer review reports and author responses from that submission.

Round 1

Reviewer 1 Report

This manuscript presents the results of a plasma-enhanced chemical vapor deposition experiment where the authors claim they fabricate nitrogen-doped amorphous silicon carbide films. Their results of enhanced luminescence upon nitrogen addition are convincing, but the underlying description and discussion are poor. Practically no details are given for the experiment, and the characterization techniques are clearly insufficient to assess the composition and structure of the resulting films. I would assume that AFM, XPS and XRD analyses would be of help. A lot more work and insight would be needed to make this work worthy of publication in an international scientific journal.

Please find below some detailed comments:

In line 45, the term “sccm” should be defined the first time that it is employed.

Too little detail is given in the Experimental Details section, and there is no reference to previous works either. More information must be given so that the experiment can be reproduced in other laboratories. There is no information either on the experimental methodology for the synthesis of the films or their evaluation.

Information on the morphological features of the films is absent.

The authors mark a band around 150 cm-1 in their Raman measurements, but they provide no assignment and make no reference to it in the text.

Even though it’s reasonable to expect that Si-N bonds will be formed with increasing NH3 concentration, I really don’t see how the Raman or FTIR results provide clear evidence for it.

The method to analyze the exponential decays of the photoluminescence is unusual. The authors should justify their choice of a double exponential if they later aim to obtain an “average lifetime”. I believe it is a single exponential that is adequate in this case.

In their resulting “average lifetimes” the authors do not include error bars, so it is impossible to assess whether their claimed lifetime increase is real.

Previous work on SiC has not been properly acknowledged. Only 12 references are included, and for some reason the authors only seem to be aware of previous works by Chinese authors, a clear under-representation of previous works on these systems.   

Reviewer 2 Report

In general, this manuscript is below the average articles published in Micromachines. In general for an article (rather than communication), it should be a complete story with multiple experimental pieces of evidence that can support the conclusions. A typical article size manuscript would contain at least 7-8 figures and schematics, with a regular length of around 4500-6000 words. This manuscript looks more like an experimental report without enough details. I would not recommend the acceptance of the manuscript under the current format.